# 5-HTT Deficiency in Male Mice Affects Healing and Behavior after Myocardial Infarction

**DOI:** 10.3390/jcm10143104

**Published:** 2021-07-14

**Authors:** Sandy Popp, Angelika Schmitt-Böhrer, Simon Langer, Ulrich Hofmann, Leif Hommers, Kai Schuh, Stefan Frantz, Klaus-Peter Lesch, Anna Frey

**Affiliations:** 1Comprehensive Heart Failure Center, University Hospital of Würzburg, 97078 Würzburg, Germany; sandy.popp@outlook.com (S.P.); Sim_Langer@gmx.de (S.L.); Hofmann_U2@ukw.de (U.H.); Frantz_S@ukw.de (S.F.); kplesch@mail.uni-wuerzburg.de (K.-P.L.); 2Center of Mental Health, Department of Psychiatry, Psychosomatics and Psychotherapy, Division of Molecular Psychiatry, University Hospital of Würzburg, 97080 Würzburg, Germany; 3Center of Mental Health, Department of Psychiatry, Psychosomatics and Psychotherapy, University of Würzburg, 97080 Würzburg, Germany; Schmitt_A3@ukw.de (A.S.-B.); hommers_l@ukw.de (L.H.); 4Medical Clinic and Policlinic I, University Hospital of Würzburg, 97080 Würzburg, Germany; 5Interdisciplinary Center for Clinical Research, University Hospital of Würzburg, 97080 Würzburg, Germany; 6Institute of Physiology I, University of Würzburg, 97070 Würzburg, Germany; kai.schuh@uni-wuerzburg.de; 7Department of Translational Neuroscience, School for Mental Health and Neuroscience, Maastricht University, 6229 Maastricht, The Netherlands; 8Laboratory of Psychiatric Neurobiology, Institute of Molecular Medicine, I.M. Sechenov First Moscow State Medical University, 119991 Moscow, Russia

**Keywords:** chronic heart failure, myocardial infarction, serotonin transporter deficient mice, anxiety, depression, behavior, inflammation

## Abstract

Anxiety disorders and depression are common comorbidities in cardiac patients. Mice lacking the serotonin transporter (5-HTT) exhibit increased anxiety-like behavior. However, the role of 5-HTT deficiency on cardiac aging, and on healing and remodeling processes after myocardial infarction (MI), remains unclear. Cardiological evaluation of experimentally naïve male mice revealed a mild cardiac dysfunction in ≥4-month-old 5-HTT knockout (−/−) animals. Following induction of chronic cardiac dysfunction (CCD) by MI vs. sham operation 5-HTT−/− mice with infarct sizes >30% experienced 100% mortality, while 50% of 5-HTT+/− and 37% of 5-HTT+/+ animals with large MI survived the 8-week observation period. Surviving (sham and MI < 30%) 5-HTT−/− mutants displayed reduced exploratory activity and increased anxiety-like behavior in different approach-avoidance tasks. However, CCD failed to provoke a depressive-like behavioral response in either *5-Htt* genotype. Mechanistic analyses were performed on mice 3 days post-MI. Electrocardiography, histology and FACS of inflammatory cells revealed no abnormalities. However, gene expression of inflammation-related cytokines (TGF-β, TNF-α, IL-6) and MMP-2, a protein involved in the breakdown of extracellular matrix, was significantly increased in 5-HTT−/− mice after MI. This study shows that 5-HTT deficiency leads to age-dependent cardiac dysfunction and disrupted early healing after MI probably due to alterations of inflammatory processes in mice.

## 1. Introduction

Recent literature shows that depression is an independent risk factor for cardiac events [1]. Depression results in reduced compliance regarding lifestyle factors and prescribed medication, which might aggravate the underlying cardiac disease [2] and is associated with increased mortality [3]. Furthermore, major depression and anxiety disorders are common comorbidities in patients with chronic heart failure (CHF), which could be the final common pathway of the coronary artery disease. In the Würzburg Interdisciplinary Network for Heart Failure (INH) registry depression at study entry was frequent [4], and an independent predictor of mortality in CHF patients [5].

In our previous experiments we could show that mice with ischemic CHF exhibit anhedonic behavior, decreased exploratory activity and interest in novelty as well as mild cognitive impairments, implicating a significant interaction between CHF and depressive disorders [6]. Besides, a substantial body of recent evidence suggests that psycho-emotional stress hastens the worsening of cardiac phenotypes that eventually lead to sudden death and other adverse events in certain animals models of metabolic [7], cardiovascular, neuropsychiatric [8] or neuromuscular disorders [9,10], thereby further underpinning the bidirectional crosstalk between the heart and the brain. Within this context, dysfunction of the serotonergic system represents a crucial pathophysiological mechanism mediating detrimental stress-evoked cardiac perturbations [11].

Altered brain serotonin (5-HT) homeostasis plays an important role in the development of anxiety disorders and depression. Preclinical and clinical studies have established that impaired 5-HT neurotransmission is a common hallmark in major depression and antidepressant drugs, specifically selective 5-HT reuptake inhibitors (SSRIs; e.g., citalopram), typically exert their therapeutic action via the 5-HT transporter (5-HTT; responsible for 5-HT reuptake from the extracellular space) [12]. Moreover, 5-HT seems to influence several processes that are important for healing and remodeling after a myocardial infarction (MI), e.g., via its effect on the 5-HT2B receptor [13,14] as up-regulation and stimulation of the 5-HT2B receptor within the heart leads to cardiac hypertrophy [15], such that mice lacking the 5-HT2B receptor are protected from cardiac hypertrophy. 5-HT can context-dependently induce both vasoconstriction and vasodilation [16], influences inflammatory responses and promotes formation of a temporary scar acting as a scaffold for normal tissue to be restored [13]. The role of 5-HT on cardiac function irrespective of myocardial infarction is complex and species-dependent, e.g., both hypo- and hypertension can be provoked via application of 5-HT [17]. However, in situations of chronic injury or damage 5-HT signaling can have deleterious effects and trigger maladaptive wound healing resulting in tissue fibrosis and impaired organ regeneration e.g., in the liver and in the lung [13,18]. In humans a correlation was found between plasmatic 5-HT and the degree of hypertrophy in aortic stenosis patients. In another study, higher 5-HT levels were associated with worse HF symptoms and systolic dysfunction [19].

Genetically modified mice lacking the serotonin transporter (5-HTT−/−) are an invaluable model to study the consequences of constitutively increased extracellular and decreased intracellular 5-HT levels due to lifelong 5-HT (re)uptake deficiency and altered 5-HT synthesis/turnover rates [18,20,21,22]. Tissue concentrations of 5-HT are decreased relative to wildtype control levels to 30–60% in several brain regions and to less than 10% in multiple peripheral tissues including the heart of 5-HTT−/− mice. Conversely, neither brain nor peripheral 5-HT tissue levels are affected in mutant mice with partial loss of *5-Htt* gene function (5-HTT+/− [21]. Furthermore, 5-HTT null mutants exhibit various changes at the (neuro-)biochemical level, e.g., brain region-specific up-/downregulated expression or altered function (desensitization) of the serotonin receptors 5-HT_1A_, 5-HT_1B_, 5-HT_2A_ and 5-HT_2C_ that may contribute to the animals’ complex phenotypic alterations [23,24,25,26]. As such, the 5-HTT mouse model has been extensively used to investigate the role of aberrant serotonergic neurotransmission in mood and anxiety disorders [27,28,29,30,31,32]. Cumulative evidence indicates that 5-HTT null mutants of both sexes manifest robust phenotypic abnormalities suggestive of differential stress susceptibility, inhibited exploratory locomotion, heightened fear and anxiety-like behavior, as well as social deficits [33,34,35]. Furthermore, it has been shown that deficiency of the 5-HTT leads to myocardial and valvular fibrosis together with left ventricular dysfunction and dilation in 8- to 10-week-old male mice on a mixed 129/Sv×C57BL/6 genetic background [36,37]. Conversely, development of hypoxia-induced pulmonary hypertension and corresponding vascular remodeling is markedly attenuated in this *5-Htt* mutant mouse strain [37], albeit hemodynamic parameters are not affected under normoxic conditions [36,37].

So far, it is still unclear if and how the pre-existence of a susceptible genotype and phenotype with altered 5-HT homeostasis influences early healing processes, cardiac remodeling, and behavioral changes after myocardial infarction (MI). In our present study, we tested the hypothesis whether experimentally induced chronic cardiac dysfunction (CCD) after MI leads to behavioral changes in genetically modified mice with a targeted mutation of the *5-Htt* gene and whether 5-HTT deficiency modulates early healing processes and remodeling after MI.

## 2. Materials and Methods

### 2.1. Ethics Statement

Mice were bred and maintained in the animal facility of the Center for Experimental Molecular Medicine at the University of Würzburg. All animal experiments were conducted in accordance with the Animal Protection Law (Directive of the European Parliament and of the Council of 22 September 2010 (2010/63/EU)) and have been reviewed and approved by the review board of the District Government of Lower Franconia and the University of Würzburg (approval reference number 54-2531.01-71/07 and 54-2531.01-88/13). Surgical procedures were performed under general anesthesia, and appropriate analgesia was provided to alleviate postoperative pain. Additionally, all efforts were made to minimize animal suffering and to reduce the number of animals used.

### 2.2. Animals

Experimental subjects were adult male mice carrying a targeted mutation of the serotonin transporter (*5-Htt*) gene (B6.129(Cg)-S*lc6a4^tm1Kpl^*/J; JAX stock #008355). The 5-HTT mouse line was generated as previously described [38] and had been fully backcrossed to C57BL/6J genetic background for more than 10 generations. The study cohorts comprised homozygous (5-HTT−/−) and heterozygous (5-HTT+/−) knockout mice and their corresponding wildtype (5-HTT+/+) siblings as controls. All mice were obtained from heterozygous breeding pairs, and pups were genotyped according to established protocols using ear punches to extract genomic DNA amplified by polymerase chain reaction (PCR). Subsequently, *5-Htt* genotypes were identified by gel electrophoresis of DNA-fragments of either 225 bp (5-HTT+/+), 272 bp (5-HTT−/−) or both (5-HTT+/−) [29]. Serotonin uptake is completely abolished in homozygous mice. 5-HTT null mutants are viable and fertile, and manifest pleiotropic phenotypic traits (including neurobehavioral, metabolic and cardiovascular alterations) that emerge during early adulthood and accumulate with increasing age [39,40,41]. Unless otherwise stated, mice were housed in littermate groups under controlled environmental conditions (14/10 h light/dark cycle, 21 ± 1 °C room temperature, 55 ± 5% humidity) with standard rodent chow diet (Ssniff Spezialdiäten, Soest, Germany) and water ad libitum.

### 2.3. Experimental Design

It is well documented that sex is a significant determinant for the development of cardiac phenotype in murine models [42,43]. Both epidemiologic and animal studies have suggested that female subjects undergo differing patterns of LV remodeling than males after myocardial infarction. It has been shown that during the chronic phase post-MI, males have significantly poorer LV function, more prominent dilatation and significant myocyte hypertrophy compared to females. Furthermore, a recent study demonstrated that although male and female rats developed similar mild HF post-MI, only male HF rats developed depression-like behaviors, including anhedonia and higher immobility in the sucrose preference and forced swim tests. Additionally, male but not female rats with HF showed mild cognitive impairments in a novel object recognition task and froze more than sham controls in the cued fear conditioning test [44,45]. 

Given that males exhibit more pronounced cardiac and behavioral phenotypes following myocardial infarction and decompensation to HF, all animal experiments described herein were performed in male mice.

#### 2.3.1. Experiment 1: Baseline Cardiological Evaluation

A total of 65 mice (5-HTT+/+: *n* = 18, 5-HTT+/−: *n* = 26, 5-HTT−/−: *n* = 21), 2–6 months of age, were subjected to echocardiography to evaluate baseline cardiac phenotypes. Afterwards, a subset of these mice (5-HTT+/+: *n* = 5, 5-HTT+/−: *n* = 13, 5-HTT−/−: *n* = 8) was included in the long-term study (experiment 2), while another subgroup (*n* = 6 per genotype) underwent hemodynamic and echocardiographic measurements at approximately 7 months of age. All but the mice used for long-term experiments and an additional naïve cohort of 19 mice (5-HTT+/+: *n* = 7, 5-HTT+/−: *n* = 7, 5-HTT−/−: *n* = 5) mice were sacrificed and organs (heart, lung, spleen) excised for further analyses.

#### 2.3.2. Experiment 2: Long-Term Study

Long-term experiments were performed on 106 mice (5-HTT+/+: *n* = 41, 5-HTT+/−: *n* = 43, 5-HTT−/−: *n* = 22) with an age of 2.5–4.5 months at study entry. All mice were single housed throughout the experimental period and allowed to acclimate for one week before being subjected to a sucrose preference test (SPT) to determine the animals’ preoperative hedonic state. Subsequently, mice were randomly allocated to MI (5-HTT+/+: *n* = 32, 5-HTT+/−: *n* = 31, 5-HTT−/−: *n* = 18) or sham (5-HTT+/+: *n* = 9, 5-HTT+/−: *n* = 12, 5-HTT−/−: *n* = 4) surgery and followed up for 8 weeks. Cardiac morphological and functional changes were evaluated by serial echocardiography on day 1, 21 and 56 after surgery. To assess postoperative changes in hedonic behavior, sucrose preference was measured six times at weekly intervals, with the first SPT being initiated four days after surgery. Starting four weeks after the surgical intervention, mice were subjected to a battery of approach-avoidance conflict tests, i.e., elevated plus maze (EPM), open field test (OFT), dark-lightbox (DLB) and social interaction test (SIT), to determine exploratory activity and anxiety-like behavior. In these exploration-based tasks, the mouse’s innate drive to approach new environments is in conflict with avoidance of potential threats [46]: mice naturally seek dark enclosed surroundings and generally avoid brightly illuminated, exposed and elevated places. The tests were conducted during the light phase between 10:00 and 15:00 in the aforementioned order with an inter-test interval of one week. Mice were sacrificed immediately upon terminal echocardiography and organs (heart, brain, lung, spleen) were excised for determination of organ weights, MI size, collagen content, and RNA sequencing (left ventricular myocardium). The experimental timeline is depicted in Figure 1.

#### 2.3.3. Experiment 3: Short-Term Study

Due to an unexpected poor survival of 5-HTT−/− mice after myocardial infarction, a short-term study was designed to further investigate the mechanistic background of increased early mortality in these mice. For this purpose, 108 mice (5-HTT+/+: *n* = 30, 5-HTT+/−: *n* = 41, 5-HTT−/−: *n* = 37) of 2.5–5 months of age were subjected to MI surgery and followed up for 3 days. 

### 2.4. Myocardial Infarction (MI) and Sham Operation 

MI operation under isoflurane anesthesia was performed according to the established protocol in our group [6,47]. In brief, mice were anaesthetized with isoflurane (about 2.0 vol.%), intubated and put on a mechanical small animal ventilator. After a left-sided thoracotomy, MI was induced by permanent ligation of the proximal portion of the left anterior descending (LAD) coronary artery. Buprenorphin (0.002 mg per mouse) was given i.p. for analgesia perioperatively. In parallel subgroups thoracotomy was performed to expose the heart, but no suture was made to the coronary artery (sham operation).

### 2.5. Echocardiographic Analysis 

Serial ultrasound analyses were performed on the Visual Sonics Vevo 1100 mouse echo machine with the echo transducer MS400 (Fujifilm Visualsonics, Toronto, ON, Canada) under light isoflurane (about 1.5 vol.%) anesthesia allowing spontaneous respiration [6,47] as described previously and in detail in the Appendix A. 

### 2.6. Hemodynamic Study

To perform terminal hemodynamic evaluation, animals were anesthetized with isoflurane. Left ventricular (LV) pressure curves were recorded by the MPVS-ultra single segment pressure-volume unit (ADInstruments Ltd., Oxford, UK) after catheter placement (1F polyimide mikro-tip pressure-volume catheter; ADInstruments Ltd., Oxford, UK) through the carotid artery in the LV cavity; systolic and diastolic blood pressure and heart rate measurements were obtained on catheter withdrawal in the thoracic aorta as previously described [48].

### 2.7. Electrocardiography

Serial electrocardiographic analysis before and after surgery under light isoflurane (about 1.5 vol.%) anesthesia allowing spontaneous respiration were captured at a sampling rate of 4000 points per second (ADInstruments, PowerLab 8/35 with BIOamplifier); and original electrocardiography (ECG) recordings were analyzed with the ECG-auto software (version 2.5.0.3, Emka technologies, Paris, France). The examinations were performed by a single researcher experienced in rodent electrocardiography blinded to the type of operation and genotype. Recordings lasted for 5 min and were screened for arrhythmias and standardized electrocardiographic parameters were assessed: heart rate in beats per minute (bpm), R wave to R wave duration in ms, PR interval in ms, P-wave duration in ms, QRS interval in ms, QT interval in ms, QTc in ms, JT interval in ms, Tpeak-to-Tend interval in ms, P amplitude in mV, R amplitude in mV, S amplitude in mV, T amplitude in mV, QTc Mitchell [49,50].

### 2.8. Sucrose Preference Test (SPT)

Mice were given a free choice between tap water and a 1% (*w*/*v*) sucrose solution in their home cage for 48 h. The position of the two drinking bottles was switched after 24 h to prevent confounding effects of side preference. Bottles were weighed every day to determine the volume of water and sucrose ingested. Values were averaged across the two days and sucrose preference was calculated as the percentage of sucrose intake relative to total fluid intake. Animals were weighed upon completion of each SPT to monitor body weight changes over time.

### 2.9. Exploration-Based Approach-Avoidance Conflict Tests

Animal behavior was automatically recorded and analyzed using a video tracking system (VideoMot2 V.5.76, TSE Systems, Bad Homburg, Germany) in the EPM, the OFT, the DLB and the SIT. To determine exploratory activity and anxiety-like behavior, the testing arenas were digitally divided into predefined regions of interest as specified in the following sections. Total distance traveled, movement duration and velocity were measured in the EPM, OFT and SIT [51] as indicators of spontaneous locomotor activity. The number of fecal boli deposited in the arena was scored at the end of each test. Further description of the tests has been published previously [6,47] and is further specified in the Appendix A.

### 2.10. Sample Collection, Determination of Infarct Size, Ventricular Remodeling, and Quantification of Heart Failure

After completion of the experiments, animals were anesthetized with a lethal dose of isoflurane. Hearts were removed and snap frozen in isopentane. The organs were weighed and embedded in paraffin and/or stored in liquid nitrogen. Left ventricles were cut into three transverse sections: apex, middle ring and base as previously reported [6,47]. From the middle ring, 5 µm sections were stained with picrosirius red. Infarct size (fraction of the infarcted left ventricle) was calculated as the percentage of length of circumference. Animals were categorized into groups with small (MI < 30%) and large (MI > 30%) infarct size.

### 2.11. Real-Time Quantitative Polymerase Chain Reaction (qPCR)

Transforming growth factor β (TGF-β), tumor necrosis factor α (TNF-α), interleukin (IL-6 and IL-10), matrix metalloproteinase (MMP-2, MMP-3, MMP-13), collagen (Col-1α1, Col-1α2, Col-3), secreted protein acidic and rich in cysteine (osteonectin), alpha smooth muscle actin 2 (Acta2), serotonin receptor (5HT2A and 5HT2B), and serotonin transporter (SERT) mRNA expression was quantified by qPCR from 3-day infarcted myocardium of 5-HTT+/+, +/−, and −/− animals in the short-term study. RNA isolation and qPCR procedures were performed with commercially available RNA extraction kits and cDNA transcription kits (Qiagen N.V., Hilden, Germany) and TaqMan probes (Applied biosystems, Foster City, CA USA) as reported previously [6,47]. Target gene ratios were normalized to GAPDH.

### 2.12. High-Performance Liquid Chromatography (HPLC)

Animals were sacrificed with a lethal dose of isoflurane. Hearts were quickly removed, snap frozen in liquid nitrogen and stored at −80 °C until further use. Then, tissue samples were solubilized in 19 times the volume of their weight in transmitter buffer (150 mM H_3_PO_4_, 500 mM DTPA) by homogenization using a Tissue Ruptor II from QIAGEN 3× on ice in CO_2_ atmosphere and centrifuged (13,000 rpm, 10 min, 4 °C). For the detection of the neurotransmitters serotonin (5-HT), dopamine (DA), norepinephrine (NE), epinephrine (EPI), and their respective metabolites [5-hydroxyindoleaceticacid (5-HIAA), 3,4-dihydroxyphenylacetic acid (DOPAC), homovanillic acid (HVA), and 3-methoxy-4-hydroxyphenylglycol (MHPG)] obtained supernatants (50 µL each) were injected into an Agilent 1100 HPLC system (Agilent Technologies, Waldbronn, Germany) consisting of EC 250/4.6 Nucleosil 100-3-C18 reversed-phase columns (Machery-Nagel, Düren, Germany) and an electrochemical detector (model 1640; BioRad, Munich, Germany) which was adjusted to 0.75 V. Composition of the mobile phase was 90% 0.65 mM octanesulfonic acid, pH 3.51, 10% methanol, 0.5 mM triethylamine, 0.1 mM EDTA and 0.1 M NaH_2_PO_4_. Monoamines and metabolites were quantified relative to standard injections of known amounts and tissue levels calculated as ng per g wet tissue weight. HPLC analysis was performed according to the previously described method [52].

### 2.13. Infarct Size Measurement in the Short-Term Study after Myocardial Infarction (Evans Blue/TTC-Staining)

Infarct size measurement 3 days after MI via Evans blue/TTC-staining was performed as recently described [6,47] and is described in more detail in the Appendix A.

### 2.14. Histological Evaluation of the Neutrophil Influx into the Scar

Cryosections of mouse myocardium were prepared in the standard manner [6,47], which is further presented in the Appendix A. The evaluation of neutrophils was performed semi-quantitatively by visual classification of neutrophil numbers in eight light microscopic 400-fold view fields: “0” indicates absence of high-density positive cells and “3” indicates strongest prevalence of high-density positive cells. Six animals per genotype were analyzed.

### 2.15. FACS Analysis of the Infarcted Tissue

Cell suspensions from the infarct zone from individual hearts were prepared by digestion with collagenase type 2 and protease type XIV (Sigma Aldrich, Germany) and stained according to protocols as recently described [6,47] and presented in detail in the Appendix A. Cells were analysed in FACS buffer on a BD FACSCanto II flow cytometer (BD, Heidelberg, Germany). Data were analyzed with FlowJo (TreeStar Inc., Ashland, OR, USA) software (version 7.6.5).

### 2.16. Collagen Quantification

Collagen quantification in the non-infarcted myocardial tissue, in the border zone and in the scar was performed as previously described [6,47,53]. 

### 2.17. Statistical Analysis

Statistical analyses were performed using IBM SPSS Statistics 21 (IBM Corp., Armonk, NY, USA) and GraphPad Prism 6 (GraphPad Software, La Jolla, CA, USA).

Data sets were checked for normality using the Shapiro–Wilk test and visual inspection of box plots. Homogeneity of variance across groups was assessed by Levene’s test. Comparisons between *5-Htt* genotypes (5-HTT+/+, 5-HTT+/−, 5-HTT−/−) were made by ordinary one-way analysis of variance (ANOVA) with Tukey post hoc test or Welch’s ANOVA with Games–Howell post hoc test in case of unequal variances. The nonparametric Kruskal–Wallis test was applied when appropriate. Baseline echocardiography and organ weight data (experiment 1) were analyzed by two-way ANOVA with age group (≤3 months vs. ≥4 months) and *5-Htt* genotype as between-subjects factors. Long-term cardiological and behavioral data (experiment 2) were analyzed by ordinary two-way ANOVA or three-way mixed ANOVA with group (sham, MI < 30%, MI > 30%) and *5-Htt* genotype as between-subjects factors and time (days or minutes) as within-subjects factor. Given that all 5-HTT−/− mice with MI > 30% had died (experiment 2), the type IV sum of squares method was applied to account for an unbalanced design with empty cells. Significant interactions in multifactorial ANOVAs were followed up by simple main effects tests with Sidak adjustment for multiple pairwise comparisons. Survival curves were derived by the Kaplan–Meier method and statistically compared using the log-rank (Mantel–Cox) test. Proportions were compared using contingency tables and analyzed via chi-square test. Pearson correlation analysis was performed to identify relationships among variables. All data are expressed as mean ± standard error of the mean (SEM). Significance levels are shown as # *p* < 0.1, * *p* < 0.05, ** *p* < 0.01 and *** *p* < 0.001 unless otherwise indicated.

## 3. Results

### 3.1. Experiment 1: Baseline Cardiological Evaluation

Echocardiographic assessment of left ventricular dimensions and function under baseline conditions revealed no overt differences between 5-Htt genotypes at 2–3 months of age (Figure 2A,B and Appendix A). At 4–6 months, however, a distinct cardiac phenotype emerged in 5-HTT−/− mice. In particular, these animals exhibited significant enlargement of LV end-systolic area (ESA; Figure 2A) and decline in LV fractional area change (FAC; Figure 2B) as compared to age-matched 5-HTT+/+ mice (*p* ≤ 0.018) and their younger 5-HTT−/− counterparts (*p* < 0.0001). The heart rate was comparable among 5-Htt genotypes but generally decreased in 4–6-month-old animals relative to the younger mice (*p* = 0.013; Appendix A).

Hemodynamic measurements obtained from 7-month-old mice (Appendix A) revealed decreased cardiac output (CO; Figure 2C) in 5-HTT−/− mice relative to 5-HTT+/+ littermates (*p* = 0.055). Conversely, the pressure at the peak rate of left ventricular filling (P dV/dtmax; Figure 2D) was significantly increased in 5-HTT−/− mice as compared to 5-HTT+/− (*p* = 0.012) and 5-HTT+/+ (*p* = 0.005) animals. In addition, the peak systolic ejection rate (dV/dtmin; Figure 2E) was slightly diminished in 5-HTT−/− and 5-HTT+/− relative to 5-HTT+/+ mice (*p* = 0.090 and *p* = 0.013, respectively). No differences could be obtained regarding systolic and diastolic blood pressure between the different genotypes. 

Histological quantification of left ventricular collagen content in the older age group revealed no significant difference between *5-Htt* genotypes (Figure 2F).

Evaluation of organ (heart, lung, spleen) weights and organ-to-body weight ratios (Appendix A) indicated reduced LV/BW, RV/BW and spleen/BW ratios in older vs. younger mice (*p* ≤ 0.016). 

Expression analysis of inflammation- and extracellular matrix remodeling-related genes in myocardial tissue of non-operated mice via qPCR yielded no overt differences between *5-Htt* genotypes. Correlational analysis indicated that the expression levels of TGF-β increased (r(40) = 0.848, *p* < 0.0001) and Col-1α1 decreased (r(24) = −0.636, *p* < 0.001) as a function of age, though these effects were not modulated by *5-Htt* genotype. Furthermore, serotonin receptor 5-HT2A and 5-HT2B expression levels were comparable among genotypes (Appendix A).

HPLC analysis of the monoamines NE, EPI, 5-HT, DA, their metabolites MHPG, 5-HIAA, DOPAC, HVA) as well as their turnover rates in myocardial tissue of non-operated mice revealed no significant differences between *5-Htt* genotypes (Appendix A).

### 3.2. Experiment 2: Long-Term Study

#### 3.2.1. Impaired Survival of 5-HTT−/− Mice after MI

The perioperative mortality rate within 24 h post-MI was 11.1% (*n* = 9/81), with no statistically significant difference between genotypes (5-HTT+/+: 15.6%, *n* = 5/32; 5-HTT+/−: 9.7%, *n* = 3/31; 5-HTT−/−: 5.6%, *n* = 1/18; χ^2^(2) = 1.287, *p* = 0.525). None of the sham-operated control mice died perioperatively (*n* = 0/25). The remaining 97 mice were used in the survival study and followed up for 8 weeks. In the sham group, one out of four 5-HTT−/− mice died 6 days after surgery, while no deaths occurred in 5-HTT+/− and 5-HTT+/+ mice (*n* = 0/12 and *n* = 0/9, respectively). Upon LAD ligation, the overall mortality rate (including all MI-operated animals) was lowest in 5-HTT+/− (32.1%, *n* = 9/28), intermediate in 5-HTT+/+ (44.4%, *n* = 12/27) and highest in 5-HTT−/− mice (64.7%, *n* = 11/17), with no statistically significant difference between genotypes (χ^2^(2) = 4.527, *p* = 0.104; Figure 3A). 

Intriguingly, however, stratification of MI mice into two distinct subgroups according to the heart failure defining cut-off infarct size of 30% [54] revealed a significant impact of genotype on survival of MI > 30% animals (χ^2^(2) = 7.519, *p* = 0.023; Figure 3B). Here, the 5-HTT null mutation was associated with 100% mortality by postoperative day 11. Conversely, at 8 weeks post-op the mortality rate of 5-HTT+/− and 5-HTT+/+ mice with large MI was 50% (*n* = 9/18) and 63.2% (*n* = 12/19), respectively. 

Non-surviving infarcted mice died of infarct rupture (data not shown) or acute heart failure (Appendix A), and deaths occurred predominantly within one week of surgery (5.53 ± 0.37 days).

#### 3.2.2. Infarct Size Quantification

At 8 weeks post-MI, the average MI size as measured by picrosirius red staining was significantly smaller in 5-HTT−/− (5.24 ± 3.50%) relative to 5-HTT+/− (26.68 ± 5.66%, *p* = 0.011) and 5-HTT+/+ mice (33.50 ± 6.93%, *p* = 0.005), while the latter two did not differ from each other (Figure 3C). Stratification into small (MI < 30%) and large (MI > 30%) infarct size subgroups showed that approximately half of the 5-HTT+/+ (46.7%, *n* = 7/15) and 5-HTT+/− (47.4%, *n* = 9/19) but none of the 5-HTT−/− (*n* = 0/5) mice had a large MI. Within the MI > 30% group, MI size did not differ between 5-HTT+/+ and 5-HTT+/− mice (59.03 ± 3.88% vs. 50.76 ± 3.14%, *p* = 0.116). No signs of MI were observed in sham-operated animals.

#### 3.2.3. Echocardiography and Terminal Organ Weights

Echocardiographic measurements performed on day 1 post-op yielded no overt differences between 5-Htt genotypes within the group of ligated mice that died prematurely and did not survive until experiment termination (Appendix A). Among the surviving animals, serial echocardiography showed that MI > 30% mice developed progressive LV dilation and severe chronic LV systolic dysfunction (Appendix A). In particular, LV dimensions at end-systole (ESD, ESA; Figure 4A) and end-diastole (EDD, EDA; Figure 4B) were significantly enlarged in MI > 30% mice relative to MI < 30% (*p* < 0.01) and sham (*p* < 0.0001) animals by day 1 and increased even further until week 8 post-op (all *p* < 0.00001). LV systolic function, expressed as fractional shortening (FS) and fractional area change (FAC) (Figure 4C), remained fairly stable over time in either group but was considerably decreased in MI > 30% mice as compared to MI < 30% (*p* < 0.001) and sham (*p* < 0.00001) animals throughout the entire follow-up period. 

Analysis of terminal organ-to-body weight ratios revealed significant enlargement of the LV (LV/BW, *p* < 0.0001; Figure 4D) and RV (RV/BW, *p* < 0.05) in MI > 30% mice compared to MI < 30% and sham-operated animals. Likewise, MI size was positively correlated with LV/BW (r = 0.706, *p* < 0.0001) and RV/BW (r = 0.394, *p* = 0.013). Lung/BW and spleen/BW ratios of the surviving MI mice did not differ from those of sham controls. Furthermore, organ-to-body weight ratios were comparable across *5-Htt* genotypes (Appendix A).

#### 3.2.4. Collagen Quantification

Quantification of the collagen content in the septum of all three groups revealed no significant differences between sham- and MI-operated 5-HTT+/+ mice. Within the large MI group, relative collagen content was significantly increased in 5-HTT+/− relative to 5-HTT+/+ mice in both septum (*p* = 0.010) and border zone (*p* = 0.037). No difference between *5-Htt* genotypes was observed for collagen within the infarct scar of MI > 30% mice (Appendix A).

#### 3.2.5. Body Weight, Liquid Intake and Sucrose Preference

Baseline body weights were comparable between 5-Htt genotypes (Figure 5A). Subsequently, however, 5-HTT−/− mice gained disproportionately more weight than 5-HTT+/− (*p* = 0.068) and 5-HTT+/+ (*p* = 0.008) mice, with the effect being particularly pronounced in sham-operated controls (Figure 5B). Preoperative daily liquid intake was significantly reduced in 5-HTT−/− mice compared to 5-HTT+/− (*p* = 0.017) and 5-HTT+/+ (*p* = 0.008) animals (Figure 5C; Appendix A), and this difference persisted throughout the postoperative period (*p* < 0.001; Figure 5D). Sucrose preference scores did not differ across *5-Htt* genotypes, neither before nor after surgery (Figure 5E,F). 

At the group level, MI > 30% mice exhibited significantly greater body weight loss than MI < 30% mice, *p* = 0.008) and sham controls (*p* < 0.0001) during the first week post-op (Figure 5B), followed by weight regain to baseline levels within the second week. Subsequent growth curves until experiment termination were comparable between the three groups. Analogously, the postoperative liquid intake was significantly diminished in MI > 30% relative to MI < 30% (*p* = 0.013) and sham (*p* = 0.021) animals during the first week (Figure 5D), but indistinguishable between groups thereafter. Surprisingly, LAD ligation failed to induce an anhedonia-like phenotype as the sucrose preference ratios remained stable and comparable among groups during the entire postsurgical period (Figure 5F).

Ligated mice that died within the early postoperative phase suffered the most severe weight loss along with a dramatic reduction in fluid intake and sucrose preference (Appendix A).

#### 3.2.6. Approach-Avoidance Conflict Tests 

##### Elevated Plus Maze

We detected no overt effects of *5-Htt* genotype on locomotor activity (cumulative distance traveled and total arm entries) and anxiety-related behaviors (exploration of the open vs. closed arms) in the EPM. Furthermore, though MI had generally little impact on emotionality in this test, it modulated the initial behavioral response of 5-HTT+/+ mice with large infarcts since these animals spent cumulatively less time in the ambiguous central start area than respective controls (*p* < 0.05) while exhibiting greater preference for the protected closed arms in the first minutes of exposure to the maze (Appendix A).

##### Open Field Test

5-HTT null mutants displayed significant behavioral deficits in the OFT both with respect to motor activity as well as anxiety. More specifically, the 5-HTT−/− mice covered persistently less distance (Figure 6A), were more immobile and showed strong avoidance of the more threatening, unprotected center area (Figure 6B) as compared to their 5-HTT+/− (*p* < 0.01) and 5-HTT+/+ (*p* < 0.05) counterparts. Moreover, MI affected OF behavior in a time- and MI size-dependent manner. In detail, MI < 30% mice exhibited greater initial activity (Figure 6A) and explored the aversive central square more thoroughly (Figure 6B) than respective sham controls (*p* < 0.05), thereby indicating decreased anxiety-like behavior. Although no overt behavioral phenotype was observed in MI > 30% mice, the number of fecal boli deposited (a measure of stress and emotionality) was significantly decreased in these animals relative to the other two groups (*p* < 0.05; Figure 6C; Appendix A).

##### Dark-Light Box 

The DLB elicited a strong neophobic response in 5-HTT−/− mice as characterized by their severe hesitation to explore the large, brightly illuminated compartment while preferring to stay in the protected dark chamber. Only 50% of the null mutants entered the light box within the 10 min (Figure 6D). Hence, the cumulative distance covered (Figure 6E) and stay time (Figure 6F) in the bright section were significantly reduced in 5-HTT−/− mice relative to the other two genotypes (*p* < 0.05). Furthermore, MI modulated risk-taking behavior in the DLB both genotype- and MI size-dependently since 5-HTT+/− mice with small (<30%) infarcts spent considerably more time in the light than their MI > 30% counterparts and compared to the other two *5-Htt* genotypes within the MI < 30% group (*p* < 0.05; Figure 6F; Appendix A).

##### Social Interaction Test

In the SIT, 5-HTT null mutants presented a distinct behavioral phenotype reminiscent of social anxiety. Besides their markedly attenuated exploratory activity, 5-HTT−/− mice showed a strong avoidance of the unfamiliar stimulus mouse as evidenced by refraining from approaching the interaction zone (Figure 6G,H) while staying passively in the opposite corners at the furthest distance from the social target (Figure 6I) (all *p* < 0.01 compared to the other two *5-Htt* genotypes). MI exerted no effects on social investigatory behavior (Appendix A). 

### 3.3. Experiment 3: Short-Term Study

Due to the unexpected poor survival of 5-HTT−/− mice following MI, we focused our further mechanistic analyses on the short-term period of 3 days post-MI to reveal mechanistic changes responsible for increased mortality.

#### 3.3.1. Infarct Size Measurement (Evans Blue/TTC Staining)

Myocardial area at risk (AAR) and infarct (INF) expressed as a percentage of left ventricle (LV) were comparable across *5-Htt* genotypes. Notably, however, the ratio of myocardial infarct area to area at risk (INF/AAR) was significantly increased in 5-HTT+/− mice relative to 5-HTT+/+ (*p* = 0.026) and 5-HTT−/− (*p* = 0.002) animals without any relevant differences between the latter (Figure 7A).

#### 3.3.2. Electrocardiography

Preoperative ECG recordings yielded no overt differences between *5-Htt* genotypes (Appendix A). Post-op, 5-HTT−/− mice demonstrated smaller R-wave (*p* = 0.084), S-wave (*p* = 0.069) and T-wave (*p* = 0.017) amplitudes as compared to 5-HTT+/+ mice (Figure 7B). Moreover, the JT interval was significantly prolonged in 5-HTT+/− relative to 5-HTT+/+ mice (*p* = 0.007) (data not shown). No further statistical differences between *5-Htt* genotypes were observed for the remaining ECG parameters (*p* duration, RR interval, PR interval, QRS interval, QT interval, QTc interval, TpTe interval). Importantly, there was no incidence of tachycardia or malignant arrhythmias in either genotype.

#### 3.3.3. Inflammation

It is well-known that inflammation impacts healing after MI. First, we performed histological evaluation for the presence of neutrophils within the scar zone, which revealed no significant difference between 5-HTT+/+ (1.44 ± 0.25, *n* = 6), 5-HTT+/− (1.82 ± 0.10, *n* = 6) and 5-HTT−/− (1.65 ± 0.09, *n* = 6) mice (χ^2^(2) = 2.236, *p* = 0.327). 

Further evaluation of immune cell subsets in the infarct and peri-infarct zone of 5-HTT+/+ (*n* = 3), 5-HTT+/− (*n* = 3) and 5-HTT−/− (*n* = 6) mice by FACS-analysis (Figure 7C) found no differences in the number of neutrophils (CD45 + CD11b + Ly6G+ cells, χ^2^(2) = 3.923, *p* = 0.141), monocytes (CD45 +/CD11b + _Ly6G-/F4/80 + _Ly6Chi cells, χ^2^(2) = 0.282, *p* = 0.868), and macrophages (CD45 +/ CD11b + _Ly6G-/Ly6Clow_F480+ cells, χ^2^(2) = 0.282, *p* = 0.868). 

#### 3.3.4. Gene Expression

##### Inflammation

TGF-β as a modulating protein involved in inflammation and extracellular matrix remodeling processes after MI was significantly increased by 2.1-fold and 3-fold in 5-HTT−/− mice relative to 5-HTT+/− (*p* = 0.017) and 5-HTT+/+ (*p* = 0.028) animals, respectively. The expression of TNF-α was about 1.9-fold higher in mice 5-HTT−/− mice as compared to 5-HTT+/+ littermates (*p* = 0.057). Furthermore, IL-6 expression was slightly elevated by 1.7-fold in 5-HTT−/− relative to 5-HTT+/+ mice (*p* = 0.071) (Figure 7D). The expression of further inflammation-related genes (IL-10, TH) did not differ between *5-Htt* genotypes (Appendix A). 

##### Modifications of the Extracellular Matrix

The expression of MMP-2 as a relevant gene involved in reconstruction of extracellular matrix, which enlargement is associated with poor healing, was significantly increased by 1.4-fold and 1.7-fold in 5-HTT−/− mice as compared to 5-HTT+/− (*p* = 0.082) and 5-HTT+/+ (*p* = 0.008) animals, respectively. Conversely, MMP-3 mRNA expression was significantly downregulated by 0.6-fold in 5-HTT+/− relative to 5-HTT+/+ mice (*p* = 0.019), while MMP-13 levels were about 1.5-fold higher in 5-HTT+/− compared to 5-HTT−/− mice (*p* = 0.066) (Figure 7D). The expression of other extracellular matrix remodeling-related genes (COL-1α1, COL-1α2, COL-3, SPARC, ACTA2) was comparable among *5-Htt* genotypes (Appendix A).

##### Modification of Serotonin Metabolism

The expression of the serotonin receptors 5-HT2A and 5-HT2B did not significantly differ between *5-Htt* genotypes. As expected, serotonin transporter (SERT) mRNA was not detectable in 5-HTT−/− mice (Appendix A) [37]. 

#### 3.3.5. HPLC

Analysis of monoamine (5-HT, DA, NE, EPI) and metabolite (5-HIAA, DOPAC, HVA, MHPG) concentrations in infarcted myocardial tissue yielded no significant differences between *5-Htt* genotypes at day 3 post-op (Appendix A).

## 4. Discussion

The present study was designed to investigate the effect of serotonin transporter (5-HTT) deficiency on cardiac and behavioral phenotypes using a mouse model of experimental MI. Our results show that naïve 5-HTT−/− mice are susceptible to developing age-dependent cardiomyopathy and heart failure. Furthermore, these animals revealed poor survival following MI and exhibited a distinct behavioral phenotype reminiscent of aspects of human anxiety. At the molecular level, multiple inflammation- and extracellular matrix remodeling-related genes were upregulated in the infarcted myocardium of 5-HTT−/− mice. Conversely, 5-HTT+/− mice showed no overt phenotype under baseline conditions. However, they exhibited the highest survival rates following MI surgery and engaged in greater risk-taking in selected behavioral tasks without typical signs of anhedonic behavior. 

### 4.1. Baseline Characterization

First, we set out to determine whether 5-HTT deficiency leads to structural and/or functional cardiac abnormalities under baseline conditions. Evaluation of LV dimensions and contractility by non-invasive transthoracic echocardiography revealed no overt genotypic differences in young adult (≤3 months old) mice. In mature adult (≥4 months old) animals, however, 5-HTT−/− mice demonstrated significant enlargement of LV end-systolic dimensions and diminished shortening fraction as compared to age-matched controls and younger 5-HTT deficient animals. Moreover, 5-HTT−/− mice exhibited hemodynamic alterations, including decreased cardiac output, at approximately 7 months of age. Together, these findings indicate that 5-HTT deficiency is associated with increased vulnerability to age-dependent development of LV dilatation with notable deterioration of cardiac function. 

In line with our results, Mekontso-Dessap et al. reported both dilated cardiomyopathy and myocardial wall motion abnormalities in 5-HTT mutants compared with age-matched wildtype mice [36]. Interestingly, however, their study was performed on 8- to 10-week-old animals, an age at which we did not observe any cardiac abnormalities. The delayed onset of a heart failure phenotype in our 5-HTT−/− mice could arise from differences in the genetic background strains used (pure C57BL/6J vs. mixed 129/Sv×C57BL/6). Furthermore, it has been shown that 5-HTT mutants develop myocardial and valvular fibrosis as indicated by marked collagen accumulation in the aortic leaflets as well as interstitial and perivascular regions in both ventricles [36]. We found no signs of cardiac fibrosis since the myocardial collagen content was comparable among 5-*Htt* genotypes at 7 months of age. 

### 4.2. Long-Term Study and Behavioral Changes after Myocardial Infarction 

In the second experiment, mice of the three 5-*Htt* genotypes were subjected to MI or sham operation and followed up for 8 weeks.

None of the 5-HTT−/− mice with a large size of MI (>30%) survived. Only a few 5-HTT−/− mice with a small MI (<30%) survived but did not develop cardiac dysfunction. These results suggest that 5-HTT deficiency is associated with an increased risk for mortality after acute MI. Conversely, the *5-Htt*+/− genotype seemed to exert a protective effect since the survival rate was slightly increased in these mice as compared to the intermediate survival rate of 5-HTT+/+ mice. In our previous experiments we recently showed that early treatment with the selective serotonin reuptake inhibitor (SSRI) citalopram, which leads to blockade of the 5-HTT, during the healing phase of MI significantly increased the risk for mortality due to left ventricular rupture [47]. The complete lack of a functional 5-HTT in 5-HTT−/− mice might lead to similar disturbances of healing processes after MI as citalopram-induced inhibition of 5-HTT. Conversely, no such deleterious effect was observed when citalopram treatment was initiated 7 days after surgery, i.e., after the termination of the early healing phase. Furthermore, Noorlander et al. found that prenatal exposure to the SSRI fluoxetine via placental transfer dramatically increased the mortality rate of the mice, which died of severe heart failure caused by dilated cardiomyopathy [55]. 

We previously showed that heart failure due to experimental MI in male C57BL/6N mice induced depression-like behavior, as indicated by a persistent anhedonic-like state (diminished sucrose preference), reduced exploratory activity and preference for novelty in the EPM and OFT, and impaired cognitive abilities in the object recognition task [6]. Contrarily, in our present experiments we detected neither signs of anhedonia nor any gross behavioral abnormalities in MI > 30% mice in approach-avoidance conflict tests except for a shift towards slightly heightened fear and cautious behavior in the EPM. The discrepancies between the present findings and our previous work [6] as well as those of others could be explained by methodological differences such as the time of initiation of behavioral assessments, the length of the inter-test interval, the order in which behavioral tests were performed, and other sensitive factors such as test duration, arena size, noise, illumination and so forth. Moreover, inconsistencies might arise from a divergence of behavioral performance and stress vulnerability among C57BL/6 substrains, with B6N mice (previous results) being generally more fearful and stress-susceptible than B6J (present work) mice [56,57], thus B6N vs. B6J mice might bear an increased risk for developing depression- and/or anxiety-like behavior upon MI.

In a recent publication, however, Bruns and coworkers analyzed the potential sequence of events with respect to different depression aspects upon heart failure in C57BL/6 mice [58]. Motility, exploration, and anxiety-like behavior were unaffected three weeks post-MI. The authors also found no evidence for a behavioral shift towards anhedonia (sucrose preference test) or despair (tail-suspension and Porsolt forced swim test, respectively), although they uncovered a subtle predisposition towards depressive-like behavior in an intermediate subpopulation at risk for learned helplessness after HF. Regardless of this subpopulation of mice with a subtle predisposition towards depressive-like behavior revealed by Burns and coworkers, their findings are consistent with our results described herein. 

Given that all 5-HTT−/− mice with large infarcts died before behavioral testing, MI < 30% animals were also included in all analyses. Interestingly, this group showed increased novelty-seeking and risk-taking behavior in a genotype- and test-specific manner. In the OFT, we detected a mild anxiolytic-like effect in all MI < 30% mice. Conversely, in the DLB and EPM only 5-HTT+/− mice with small infarcts appeared to be less anxious than controls. 

Moreover, surviving 5-HTT−/− mice (sham and MI < 30%) exhibited a distinct phenotype reminiscent of increased depression- and anxiety-like behavior as reflected by decreased total distance traveled in the OFT and SIT and cumulatively less stay time in the unprotected areas of the OF and DLB as well as in the social stimulus zone in the SIT. Taken together, 5-HTT deficient mice displayed severely depressed locomotor activity and strong avoidance responses when presented with novel environmental stimuli. Unfortunately, it is difficult to dissociate impaired locomotor activity from anxiety-like behavior in these ethological tests since they rely on the animals’ innate motivation to explore an unfamiliar environment. Thus, the increased avoidance behavior in 5-HTT−/− mice could be a consequence of their low exploratory drive which, in turn, is indicative of depression- rather than anxiety-like behavior. However, we found no evidence for anhedonia, i.e., a reduced ability to experience pleasure [59], since 5-HTT−/− mice similarly preferred the sweetened solution over water in the sucrose preference test. The behavioral profile observed in 5-HTT deficient mice is in agreement with previous reports, showing hypolocomotion and increased anxiety-like behavior but no anhedonia in 5-HTT mutants [28,33,60].

### 4.3. Mechanistic Considerations in the Short-Term Study

To further evaluate the mechanistic background of the poor survival rates observed in 5-HTT−/− mice, we performed several additional experiments up to three days after MI. The size of MI did not differ within the *5-Htt* genotypes.

Intake of SSRI in humans is associated with abnormalities on ECG and a tendency to increased rates of malignant arrhythmias [61,62], however no relevant pathologies except for differences in amplitudes of R-, S-und T-waves were found in 5-HTT−/− mice. Thus, the increased mortality of 5-HTT−/− mice is not attributable to malignant arrhythmias. 

Sympathetic system plays an important role in control of cardiac function [63]. Our further hypothesis was, if overshooting sympathetic regulation could have been present in 5-HTT−/− mice being responsible for impaired survival. However, the levels of monoamines NE, EPI, 5-HT and DA as well as their respective metabolites did not differ in HPLC analysis within the infarcted myocardium in three 5-*Htt* genotypes. 

In our previous study we showed that MMP-13 expression was significantly increased in citalopram-treated mice after MI. Pre-treatment with MMP inhibitor PD 166793 partially restored the deleterious effects of citalopram treatment in the acute phase after MI, prevented left ventricular rupture and improved survival [47]. Matrix metalloproteinases (MMP) are enzymes of the extracellular matrix capable of degrading all kinds of extracellular matrix proteins, but also processing a number of bioactive molecules [64]. Overexpression of several MMP subtypes was associated with disturbed early healing processed after MI [65]. MMP-2, MMP-3 and MMP-13 were differentially expressed in a genotype-dependent manner in our study: MMP-2 was significantly upregulated in 5-HTT−/− mice; conversely, MMP-3 was significantly downregulated in 5-HTT+/− relative to 5-HTT+/+ and MMP-13 was upregulated in 5-HTT+/− as compared to 5-HTT−/− mice. Overexpression of MMP-2 in 5-HTT−/− mice could be responsible for further scar changes which lead to the disturbance of early healing processes and impaired survival after MI. 

Matsumura et al. demonstrated the MMP-2 generated fragments of laminin inhibited migration of macrophages into the inflammatory region and resulted in delayed wound healing after MI [66]. Thus, we focused our analysis on the changes of the inflammatory system in *5-Htt* genotypes after MI. However, neither histological differences in the number of neutrophils within the scar zone nor alterations in the numbers of neutrophils, monocytes and macrophages as measured by FACS analysis was revealed between the 5-*Htt* genotypes within the first 3 days after MI. 

Nevertheless, the expression of inflammation-related genes TGF-β, TNF-α and IL-6 was upregulated in 5-HTT−/− mice. TNF-α and IL-6 are both classic pro-inflammatory cytokines involved in early healing processes after MI. TNF-α −/− mice or wild-type mice treated with anti-TNF antibody exhibit smaller infarcts, attenuated leukocyte infiltration, and lower expression of chemokines and adhesion molecules after ischemia-reperfusion injury [67]. Mutant mice with augmented activation of the common receptor subunit for the IL-6 cytokine exhibit adverse remodeling and heightened myocardial inflammation [68]. Thus, upregulation both of TNF-α and IL-6 could be associated with impaired early healing processes in 5-HTT−/− mice. 

Several members of the TGF-β family have been implicated in negative regulation of the inflammatory reaction [69]. Unfortunately, investigation of the role of these mediators in post-infarction inflammation and repair has been hampered by the complex biology of their regulation and activation, by their pleiotropic and context-dependent actions on all cell types involved in infarct healing, and by the complexity of their downstream signaling effectors. Neutralization experiments using gene therapy with the extracellular domain of the type II TGF-β receptor in a model of MI suggested that early inhibition may worsen dysfunction accentuating the inflammatory response [70]. On the other hand, mice with TGF-β receptor-coupled signaling genetically suppressed only in cardiac myocytes were protected against early-onset mortality due to wall rupture [71], which could possibly in part explain the deleterious effect of TGF-β overexpression in 5-HTT−/− mice in our experiments.

### 4.4. Limitations

Some limitations of our study need to be addressed. 

Due to established echocardiographic criteria, we could monitor CCD development in young adult male 5-HTT mice with and without experimentally induced MI, and shed light on the consequences of 5-HTT deficiency on post-MI anxiety- and depression-like behaviors by means of ethologically relevant paradigms. In the general population, however, depression is more common in women than men, and prevalence rates are also higher in female CHF patients, albeit male sex predicts poorer outcomes [72,73]. Thus, further studies including female cohorts are warranted to decipher potential sex-specific (e.g., estrogen-driven) mechanisms regulating MI-induced cardiac remodeling and its impact on behavior due to defective 5-HTT neurotransmission. Besides, the 5-HT signaling system is implicated in a wide range of aging-related metabolic, cardiovascular [74] and behavioral deteriorations. As such, additional research on middle-aged (10–12-month-old) or even reproductively senescent (18- to 24-month-old) mice is needed to fully elucidate the modulatory role of compromised 5-HTT function on cardiac disease progression during aging. 

Mechanistically, inflammation-related genes were found to be strongly upregulated in the hearts of 5-HTT−/− mice after MI. However, other relevant biomarkers for myocardial injury like cardiac troponin or natriuretic peptides have not been analyzed in the present study. Apart from the heart, markers for neuroinflammation are known to be increased in mice after MI [75] and also in 5-HTT−/− mice without any further intervention. In future studies, neuroinflammatory gene expression alterations associated with MI should be assessed in relevant brain regions of *5-Htt* mutant mice. 

Finally, therapeutic intervention to further modulate inflammation or extracellular matrix, e.g., pretreatment with an MMP inhibitor as performed by Frey and coworkers [47], to prevent increased mortality of 5-HTT−/− mice after MI has not been performed. We decided against it mainly due to a missing clinical and translational relevance as far as a complete 5-HTT knockout genotype does not exist in humans.

## 5. Conclusions

This study shows that 5-HTT deficiency leads to age-dependent cardiac dysfunction and disrupted early healing after myocardial infarction, which was associated with increased local inflammation and *MMP-2* expression during the early stage after MI. The baseline behavioral profile observed in 5-HTT deficient mice is in agreement with previous reports, showing hypolocomotion and increased anxiety-like behavior but no anhedonia in 5-HTT mutants. MI results in small neurobehavioral changes in different approach-avoidance tasks, but fails to provoke a depressive-like behavioral response in either *5-Htt* genotype in the SPT.

## Figures and Tables

**Figure 1 jcm-10-03104-f001:**
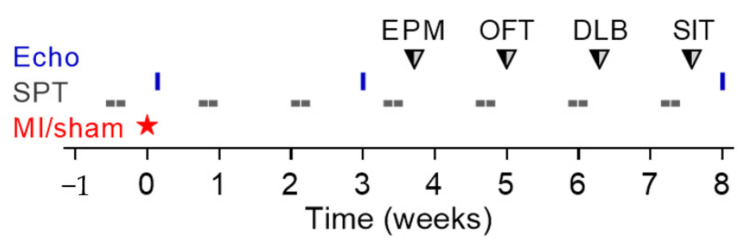
Experimental timeline of the long-term study. MI, myocardial infarction; Echo, echocardiography; SPT, sucrose preference test; EPM, elevated plus maze; OFT, open field test; DLB, dark-light box; SIT, social interaction test.

**Figure 2 jcm-10-03104-f002:**
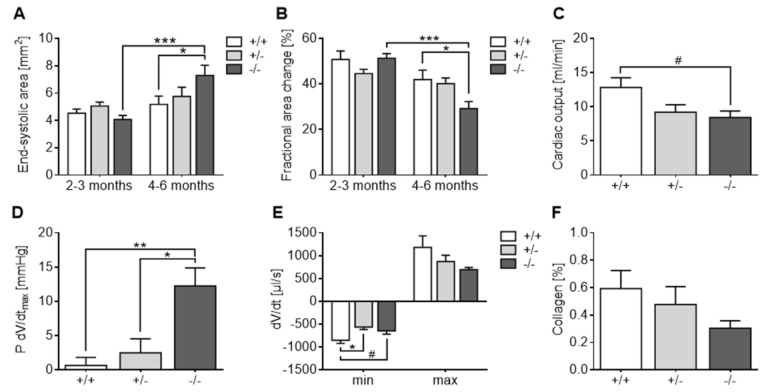
Age-dependent development cardiac dysfunction in 5-HTT deficient (−/−) mice. (**A**,**B**) Echocardiographic assessment of left ventricular dimensions and function revealed a significant increase in end-systolic area (**A**) and decline in fractional area change (**B**) in 4–6-month-old 5-HTT−/− mice relative to age-matched controls and 2–3-month-old 5-HTT−/− mice (*n* = 8–18 per age group and genotype). (**C**–**E**) Hemodynamic measurements from 7-month-old mice demonstrated slightly diminished cardiac output (**C**), increased pressure at the peak rate of left ventricular filling (**D**) and decreased peak systolic ejection rate (**E**) in 5-HTT−/− mice as compared to 5-HTT+/+ littermates (*n* = 5–6/genotype). (**F**) Myocardial collagen content did not differ between 5-Htt genotypes (*n* = 6/genotype). Data are shown as mean ± standard error of the mean (SEM). # *p* < 0.1, * *p* < 0.05, ** *p* < 0.01 and *** *p* < 0.001; two-way analysis of variance (ANOVA) followed by Sidak multiple comparisons test (**A**,**B**) or one-way ANOVA with Tukey post hoc test (**C**–**F**).

**Figure 3 jcm-10-03104-f003:**
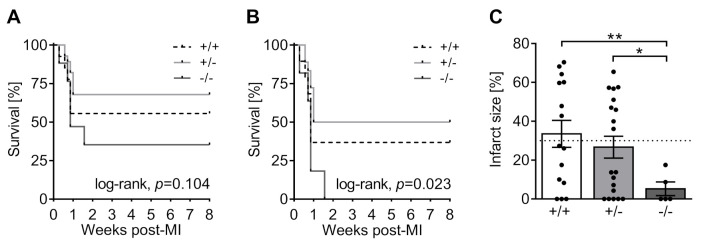
Survival of 5-HTT−/− mice is severely impaired after MI. (**A**,**B**) Kaplan–Meier survival curves of (**A**) all mice subjected to MI surgery show no statistically significant difference between 5-Htt genotypes, but (**B**) among those animals with large infarcts survival is severely impaired in 5-HTT−/− mice. (**C**) Histological quantification of the infarct size (fraction of the infarcted left ventricle) 8 weeks after surgery showed that none of the surviving 5-HTT−/− mice had a large (>30%) MI. Dashed line indicates the cut-off value of 30% for stratification into groups with small (<30%) and large (>30%) MI size. Values are shown as mean ± SEM. (**A**,**B**) log-rank test, (**C**) Welch’s ANOVA with Games–Howell post hoc test. * *p* < 0.05, ** *p* < 0.01.

**Figure 4 jcm-10-03104-f004:**
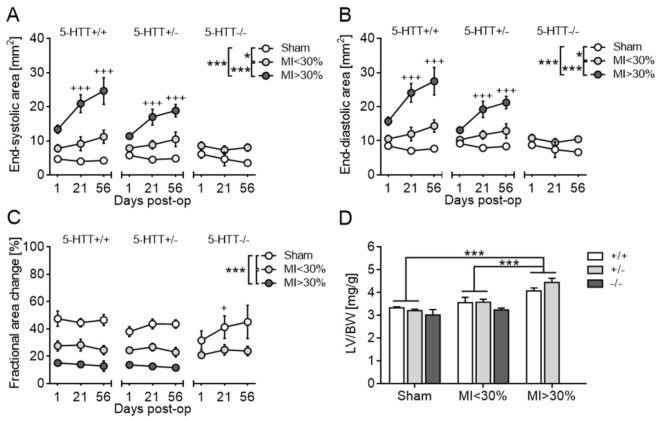
Development of left ventricular failure and hypertrophy after MI in mice. (**A**–**C**) Assessment of left ventricular dimensions and function by serial echocardiography on postoperative days 1, 21 and 56. Irrespective of *5-Htt* genotype, MI mice exhibited a time- and infarct size-dependent increase in end-systolic area (**A**) and end-diastolic area (**B**) and a persistent reduction in fractional area change (**C**) as compared to sham controls, indicating progressive LV dilatation and dysfunction due to heart failure. (**D**) Analysis of left ventricle-to-body weight (LV/BW) ratio as an index of cardiac hypertrophy revealed significant enlargement in MI > 30% mice relative to MI < 30% and sham animals at 8 weeks post-op. Data are shown as mean ± SEM. * *p* < 0.05 and *** *p* < 0.001 for comparisons between groups; + *p* < 0.05 and +++ *p* < 0.001 compared to day 1; (**A**–**C**) three-way mixed ANOVA and (**D**) ordinary two-way ANOVA followed by Sidak or Tukey multiple comparisons test.

**Figure 5 jcm-10-03104-f005:**
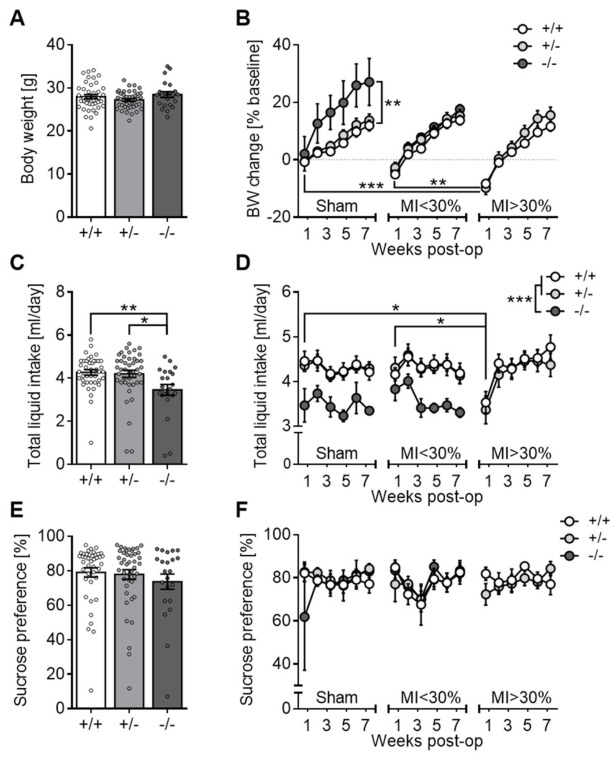
Body weight, liquid intake and sucrose preference before and after surgery. Presurgical body weights were comparable among genotypes (**A**). Postoperatively, sham-operated 5-HTT−/− mice gained more weight than respective controls and mice with large infarcts (MI > 30%) exhibited greater weight loss than sham and MI < 30% mice in the first week post-op (**B**). 5-HTT null mutants drank less than the other two genotypes both before (**C**) and after surgery (**D**), and liquid intake was significantly diminished in MI > 30% mice at week 1 post-op (**D**). Sucrose preference was comparable among genotypes at baseline (**E**) and did not significantly change postoperatively, neither across genotypes nor between sham vs. MI mice (**F**). Data are shown as mean ± SEM. * *p* < 0.05, ** *p* < 0.01, *** *p* < 0.001. (**A**,**C**,**E**) one-way ANOVA followed Tukey post hoc test; (**B**,**D**,**F**) three-way mixed ANOVA followed by Sidak multiple comparisons test.

**Figure 6 jcm-10-03104-f006:**
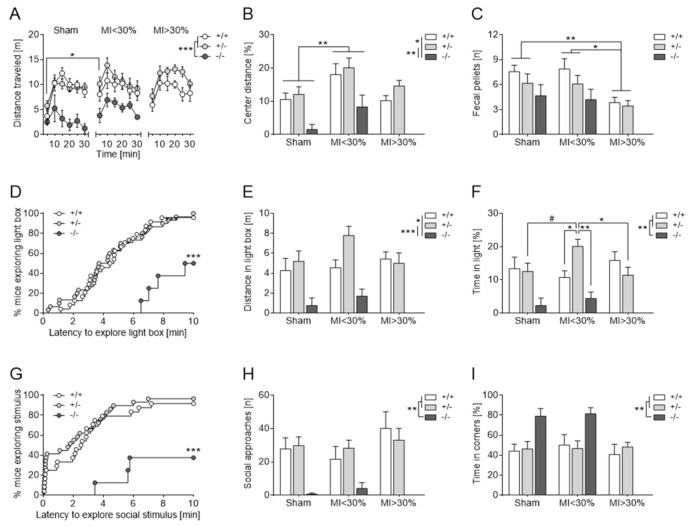
Distinct behavioral phenotype of 5-HTT deficient mice in multiple approach-avoidance conflict tests. (**A**–**C**) During the 30-min open field test (OFT), 5-HTT−/− traveled significantly shorter distances than the other two *5-Htt* genotypes in the entire arena (**A**) and in the center (**B**). Furthermore, MI < 30% were more active than sham animals during the first 2 min of the OFT (**A**) and covered cumulatively longer distances in the center of the arena as compared to sham controls (**B**). Interestingly, MI > 30% deposited less fecal pellets in the OF than MI < 30% and sham animals (**C**). (**D**,**E**) In the dark-light box (DLB), half of the 5-HTT−/− mice refrained from exploring the bright section (**D**). Furthermore, 5-HTT−/− mice covered shorter distances (**E**) and spent cumulatively less time in the light chamber (**F**) as compared to 5-HTT+/− and 5-HTT+/+ mice. Additionally, 5-HTT+/− mice with small infarcts (MI < 30%) spent significantly more time in the light compartment of the DLB than 5-HTT+/− mice of the other two groups as well as 5-HTT+/+ and 5-HTT−/− mice with MI < 30% (**F**). (**G**–**I**) In the social interaction test (SIT), only 3 out of 8 5-HTT−/− mice explored the unfamiliar stimulus mouse at least once (**G**). The number of social approaches was significantly decreased (**H**) while the cumulative time in opposite corners was dramatically increased (I) in 5-HTT null mutants compared to the other two *5-Htt* genotypes. Data are shown as mean ± SEM. # *p* < 0.1, * *p* < 0.05, ** *p* < 0.01 and *** *p* < 0.001; three-way mixed ANOVA followed by Sidak or Tukey multiple comparisons test; (**D**,**G**) log-rank test.

**Figure 7 jcm-10-03104-f007:**
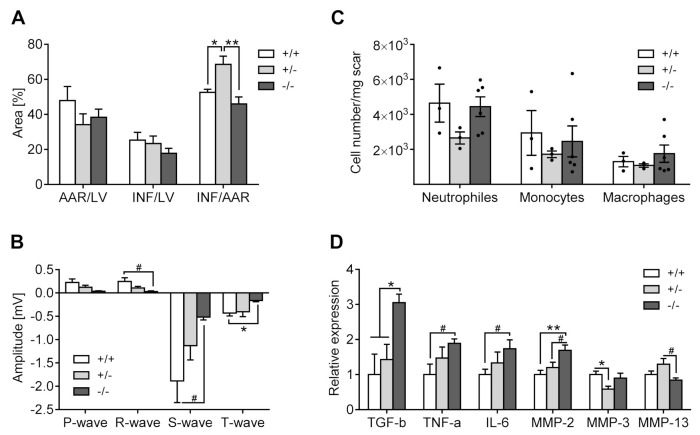
Infarct size quantification, electrocardiographic recordings, FACS and gene expression analysis in infarcted mice at day 3 post-op. (**A**) Evans blue/TTC staining revealed a significantly increased infarct-to-area at risk (INF/AAR) ratio in 5-HTT+/− compared to 5-HTT+/+ and 5-HTT−/− mice. The fraction of AAR and INF relative to left ventricle (LV) was comparable among *5-Htt* genotypes. (**B**) Postoperative electrocardiography (ECG) recordings showed significantly smaller R-, S- and T-wave amplitudes in 5-HTT−/− mice as compared to 5-HTT+/+ littermates. (**C**) Evaluation of immune cell subsets in the infarct area by FACS analysis revealed no differences between *5-Htt* genotypes. (**D**) Gene expression analysis in the infarcted myocardium revealed significant upregulation of transforming growth factor β (TGF- β), tumor necrosis factor α (TNF-α), interleukin 6 (IL-6) and matrix metalloproteinase 2 (MMP2) in 5-HTT−/− relative to 5-HTT+/+ mice. Furthermore, matrix metalloproteinase 3 (MMP-3) was downregulated and matrix metalloproteinase 13 (MMP-13) upregulated in 5-HTT+/− mice as compared to 5-HTT+/+ and 5-HTT−/− mice, respectively. Data are shown as mean ± SEM. # *p* < 0.1, * *p* < 0.05 and ** *p* < 0.01; (**A**) ordinary one-way ANOVA with Tukey post hoc test, (**B**,**C**) Welch’s ANOVA with Games–Howell post hoc test.

## Data Availability

The datasets used and analyzed for the current study are available from the corresponding author on request.

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
