# Peer review of "5-HTT Deficiency in Male Mice Affects Healing and Behavior after Myocardial Infarction"

_jcm, 2021, doi:10.3390/jcm10143104_

Round 1

Reviewer 1 Report

The manuscript entitled 5-HTT deficiency in male mice affects healing and behavior after myocardial infarction is characterized by interesting and useful insights for understanding how serotonin and its transporter SERT are able not only to influence responses at central level, but also at cardiac one.

The work is well designed from an experimental point of view and abounds with many research methodologies that enrich its content.

However, the research has some limitations that in my opinion need to be overcome.

Major Concerns

Despite the massive amount of experiments performed, it remains to be determined why naïve 5-HTT-/- mice are more prone to develop age-dependent cardiomyopathy and heart failure. Echocardiographic and hemodynamic data are shown to support the thesis, however as the authors pointed out, no LV fibrosis, structural alterations and remodeling is present in the heart of the 5-HTT-/- mice. Furthermore, the hemodynamic data do not clearly support the thesis of an intrinsic myocardial impairment, indeed if it’s true that CO (measure not load independent and affected by HR that is diminished in the mutants) is decreased, dP/dt max is totally unchanged and what is more no measures afterload independent are shown (i.e. ESPVR and PRSW). Conversely the most affected parameter among the hemodynamic ones, P at dV/dt max (with an enormous increase observed in the total KO mice), seems to indicate an afterload problem (supported also by the trend observed in EA).

In light of this, I advise the authors to delve into the role of systemic blood pressure. Indeed, many works suggest a crucial role of serotonin (Watts et al. 2012) and of the transporter SERT (Patrick Davis et al. 2011) in regulating systemic blood pressure. A role of hypertension could also explain the enormous increase in mortality after MI, with no differences in the infarcted area observed among genotypes.

I also advise the authors to deepen the RV condition (for example measuring collagen levels), indeed Crona (2009) showed how the same mice are more vulnerable to pulmonary hypertension.

Minor Concerns

Introduction is rambling and too long. I suggest the authors shorten and center it on the main focus of the manuscript that is myocardial infarction and possible repercussion of it on behavior. I also think it is worth expanding in the introduction the theme of the bidirectional heart-brain relationship, with more recent works such as Razzoli et al 2020 and Agrimi et al 2020

Another minor comment concerns the emphasis placed by the authors on a causal link between inflammatory markers and mortality in KO mice. I would advise the authors to soften the terminology by emphasizing more a correlation between phenomena, in fact there is no evidence of a causal link in the experiments shown.

Watts SW, Morrison SF, Davis RP, Barman SM. Serotonin and blood pressure regulation. Pharmacol Rev. 2012;64(2):359-388. doi:10.1124/pr.111.004697

Patrick Davis R, Linder AE, Watts SW. Lack of the serotonin transporter (SERT) reduces the ability of 5-hydroxytryptamine to lower blood pressure. Naunyn Schmiedebergs Arch Pharmacol. 2011 May;383(5):543-6. doi: 10.1007/s00210-011-0622-1. Epub 2011 Mar 30. PMID: 21448568; PMCID: PMC3097417.

Crona, D., Harral, J., Adnot, S. et al. Gene expression in lungs of mice lacking the 5-hydroxytryptamine transporter gene. BMC Pulm Med 9, 19 (2009). https://doi.org/10.1186/1471-2466-9-19

Razzoli M, Lindsay A, Law ML, Chamberlain CM, Southern WM, Berg M, Osborn J, Engeland WC, Metzger JM, Ervasti JM, Bartolomucci A. Social stress is lethal in the mdx model of Duchenne muscular dystrophy. EBioMedicine. 2020 May;55:102700. doi: 10.1016/j.ebiom.2020.102700. Epub 2020 Mar 16. PMID: 32192914; PMCID: PMC7251247.

Agrimi J, Scalco A, Agafonova J, Williams Iii L, Pansari N, Keceli G, Jun S, Wang N, Mastorci F, Tichnell C, Murray B, James CA, Calkins H, Zaglia T, Paolocci N, Chelko SP. Psychosocial Stress Hastens Disease Progression and Sudden Death in Mice with Arrhythmogenic Cardiomyopathy. J Clin Med. 2020 Nov 24;9(12):3804. doi: 10.3390/jcm9123804. PMID: 33255451; PMCID: PMC7761318.

Author Response

Answer to Review 1 comments / jcm-1266884

Reviewer #1:
The manuscript entitled 5-HTT deficiency in male mice affects healing and behavior after myocardial infarction is characterized by interesting and useful insights for understanding how serotonin and its transporter SERT are able not only to influence responses at central level, but also at cardiac one.

The work is well designed from an experimental point of view and abounds with many research methodologies that enrich its content.

However, the research has some limitations that in my opinion need to be overcome.

Despite the massive amount of experiments performed, it remains to be determined why naïve 5-HTT-/- mice are more prone to develop age-dependent cardiomyopathy and heart failure. Echocardiographic and hemodynamic data are shown to support the thesis, however as the authors pointed out, no LV fibrosis, structural alterations and remodeling is present in the heart of the 5-HTT-/- mice. Furthermore, the hemodynamic data do not clearly support the thesis of an intrinsic myocardial impairment, indeed if it’s true that CO (measure not load independent and affected by HR that is diminished in the mutants) is decreased, dP/dt max is totally unchanged and what is more no measures afterload independent are shown (i.e. ESPVR and PRSW). Conversely the most affected parameter among the hemodynamic ones, P at dV/dt max (with an enormous increase observed in the total KO mice), seems to indicate an afterload problem (supported also by the trend observed in EA).

In light of this, I advise the authors to delve into the role of systemic blood pressure. Indeed, many works suggest a crucial role of serotonin (Watts et al. 2012) and of the transporter SERT (Patrick Davis et al. 2011) in regulating systemic blood pressure. A role of hypertension could also explain the enormous increase in mortality after MI, with no differences in the infarcted area observed among genotypes.

I also advise the authors to deepen the RV condition (for example measuring collagen levels), indeed Crona (2009) showed how the same mice are more vulnerable to pulmonary hypertension.

We thank the reviewer for the important remark that indeed the pathophysiology of the cardiac dysfunction in older 5-HTT -/- mice has not so far been completely revealed by our experiments. However, the focus of our manuscript was to dissect the role of SERT deficiency in modulating the impairment of left ventricular  function and remodeling processes after myocardial infarction. Unfortunately, due to the established protocol for LV-hemodynamics and the established echo protocol for LV functioning, the role of RV function has not been further analyzed in our mice model. Under control conditions, both 5-HTT deficient mice and rats display normal systemic blood pressure (Homberg et al. 2006; Mekontso-Dessap et al. 2009).  As revealed by Crona et al in 2009 (BMC Pulmonary Medicine 2009, 9:19. doi:10.1186/1471-2466-9-19) mice with a specific gene knockout for 5-HTT (5-HTT-/-) are also somewhat protected against hypoxic pulmonary hypertension, with slightly lower right ventricular systolic pressure (RVSP) in response to chronic hypoxia. In line with this, it has been shown that serotonin transporter inhibitors protect against hypoxic pulmonary hypertension, albeit conflicting reports also showed that the use of antidepressants such as SSRIs was associated with increased mortality and greater risk of clinical worsening. On the other hand, several association studies found an increased PAH risk in 5-HTT L/L-allele carriers. This was further confirmed in mutant mice with ubiquitous or smooth muscle-specific overexpression of SERT (MacLean 2017 and references therein). Thus, the role of serotonin and its transporter in regulating systemic blood pressure and vascular remodeling is highly complex and context dependent.

However, based on previous reports on the 5-HTT KO mouse model, we did not necessarily expect that the dysfunction of the RV could further explain the dysfunction of LV, which was found by us. We have included the information regarding the influence of 5-HTT-/- on RV-pressure in the introduction part of the manuscript.

Unfortunately, we are not able to perform further analysis of the RV within the context of the presented study, since it would take several months to breed the mice, wait for aging and perform the experiments. We won’t be able to include those data in the current manuscript so far, but are happy to take a closer look on the RV function in the upcoming studies.

PAGE 2, LINE 103: Furthermore, it has been shown that deficiency of the 5-HTT leads to myocardial and valvular fibrosis together with left ventricular dysfunction and dilation in 8-to-10-week-old male mice on a mixed 129/Sv×C57BL/6 genetic background [43, 44]. Conversely, development of hypoxia-induced pulmonary hypertension and corresponding vascular remodeling is markedly attenuated in this 5-Htt mutant mouse strain [44], albeit hemodynamic parameters are not affected under normoxic conditions [43, 44].

MacLean M (Mandy) R. The serotonin hypothesis in pulmonary hypertension revisited: targets for novel therapies (2017 Grover Conference Series). Pulmonary Circulation. April 2018. doi:10.1177/2045894018759125

Homberg J, Mudde J, Braam B, Ellenbroek B, Cuppen E, Joles JA. Blood pressure in mutant rats lacking the 5-hydroxytryptamine transporter. Hypertension. 2006 Dec;48(6):e115-6; author reply e117. doi: 10.1161/01.HYP.0000246306.61289.d8. Epub 2006 Oct 9. PMID: 17030677.

Mekontso-Dessap A, Brouri F, Pascal O, Lechat P, Hanoun N, Lanfumey L, Seif I, Benhaiem-Sigaux N, Kirsch M, Hamon M, Adnot S, Eddahibi S. Deficiency of the 5-hydroxytryptamine transporter gene leads to cardiac fibrosis and valvulopathy in mice. Circulation. 2006 Jan 3;113(1):81-9. doi: 10.1161/CIRCULATIONAHA.105.554667. Epub 2005 Dec 27. PMID: 16380550.

Introduction is rambling and too long. I suggest the authors shorten and center it on the main focus of the manuscript that is myocardial infarction and possible repercussion of it on behavior. I also think it is worth expanding in the introduction the theme of the bidirectional heart-brain relationship, with more recent works such as Razzoli et al 2020 and Agrimi et al 2020

Due to the valuable comment of the reviewers, we refined and shortened the introduction and referenced recently published data regarding the deterioration of the cardiac phenotype in mice with preexisting morbidity after induction of experimental stress and the role of 5-HTT deficiency on the pulmonary pressure (REVIEWER 1). Furthermore, we added the information, e.g. on the role of 5-HT on cardiac function and dysfunction in experimental models and in humans, and the serotonin concentrations due to 5-HTT deficiency (REVIEWER 2).

PAGE 2, LINE 56: In our previous experiments we could show that mice with ischemic CHF exhibit anhedonic behavior, decreased exploratory activity and interest in novelty as well as mild cognitive impairments, implicating a significant interaction between CHF and depressive disorders [6]. Besides, a substantial body of recent evidence suggests that psycho-emotional stress hastens the worsening of cardiac phenotypes that eventually lead to sudden death and other adverse events in certain animals models of metabolic [7], cardiovascular, neuropsychiatric [8] or neuromuscular disorders [9, 10], thereby further underpinning the bidirectional crosstalk between the heart and the brain. Within this context, dysfunction of the serotonergic system represents a crucial pathophysiological mechanism mediating detrimental stress-evoked cardiac perturbations [11].

Another minor comment concerns the emphasis placed by the authors on a causal link between inflammatory markers and mortality in KO mice. I would advise the authors to soften the terminology by emphasizing more a correlation between phenomena, in fact there is no evidence of a causal link in the experiments shown.

As advised by the reviewer, we have softened the wording regarding the pathophysiological role of inflammatory processes in early healing after myocardial infarction. We have re-worded the abstract and the discussion as following:

PAGE 1, LINE 38: This study shows that 5-HTT deficiency leads to age-dependent cardiomyopathy and disrupted early healing after MI probably due to alterations of inflammatory processes in mice.

PAGE 19, LINE 900: Nevertheless, the expression of inflammation-related genes TGF-β, TNF-α and IL-6 was strongly upregulated in 5-HTT -/- mice.

PAGE 19, LINE 916: On the other hand, mice with TGF-β receptor-coupled signaling genetically suppressed only in cardiac myocytes were protected against early-onset mortality due to wall rupture [63], which could possibly in part explain the deleterious effect of TGF-β overexpression in 5-HTT -/- mice in our experiments.

PAGE 20, LINE 948: Finally, therapeutic intervention to further modulate inflammation or extracellular matrix, e.g pretreatment with a MMP inhibitor as performed by Frey and coworkers [57], to prevent increased mortality of 5-HTT-/- mice after MI has not been performed.

PAGE 20, LINE 956: This study shows that 5-HTT deficiency leads to age-dependent cardiomyopathy and disrupted early healing after myocardial infarction, which was associated with most likely due to increased local inflammation and MMP-2 expression during the early stage after MI.

Reviewer 2 Report

Popp et al. 2021 studies the effect of 5HTT (serotonin transporter) in relation to aging and healing and specifically after myocardial infarction. Overall, the study is interesting and the following questions/comments need to be addressed to improve the overall significance of the study.

1- In the Introduction, it would be important to highlight the correlation between the serotonin homeostasis and the cardiac phenotype specially in aging in addition to including previous studies that reported any correlation between serotonin and the development of cardiac phenotype.

2- The authors highlighted the importance of gender on the development of depression and anxiety. However, they did not show or presented their data in terms of gender differences.

3- More description on the mouse model needs to be included in the methods section.

4- The effect of aging is not really clear in the presented experiments and the authors only chose to compare mice less than 3 months and more than 4 months, which is not enough to support their conclusions. An older age mice group at 12 10-12 months is needed to support their findings.

5-Were there any changes in the inflammatory mediators between the adult and the young mice?

6- It would be interesting to show the difference in altered serotonin expression between the het and the homo KO.

Author Response

Answer to Review 2 comments / jcm-1266884

Reviewer #2:

Popp et al. 2021 studies the effect of 5HTT (serotonin transporter) in relation to aging and healing and specifically after myocardial infarction. Overall, the study is interesting, and the following questions/comments need to be addressed to improve the overall significance of the study.

In the Introduction, it would be important to highlight the correlation between the serotonin homeostasis and the cardiac phenotype specially in aging in addition to including previous studies that reported any correlation between serotonin and the development of cardiac phenotype.

Due to the valuable comment of the reviewers, we refined and shortened the introduction and referenced recently published data regarding the deterioration of the cardiac phenotype in mice with preexisting morbidities after induction of experimental stress (REVIEWER 1). Furthermore, we added the information, e.g. on the role of 5-HT on cardiac function and dysfunction in experimental models and in humans, and the serotonin concentrations due to 5-HTT deficiency (REVIEWER 2). 

PAGE 2, LINE 59: Besides, a substantial body of recent evidence suggests that chronic psycho-emotional stress hastens the worsening of cardiac phenotypes that eventually lead to sudden death and other adverse events in certain animals models of metabolic [7], cardiovascular, neuropsychiatric [8] or neuromuscular disorders [9, 10], thereby further underpinning the bidirectional crosstalk between the heart and the brain. Within this context, dysfunction of the serotonergic system represents a crucial pathophysiological mechanism mediating detrimental stress-evoked cardiac perturbations [11].

PAGE 2, LINE 72: Moreover, 5-HT seems to influence several processes that are important for healing and remodeling after a myocardial infarction (MI), e.g. via its effect on the 5-HT2B receptor [13, 14] as up-regulation and stimulation of the 5-HT2B receptor within the heart leads to cardiac hypertrophy [15], such that mice lacking the 5-HT2B receptor are protected from cardiac hypertrophy.

PAGE 2, LINE 79: The role of 5-HT on cardiac function irrespective of myocardial infarction is complex and species-dependent, e.g. both hypo- and hypertension can be provoked via application of 5-HT [17].

PAGE 2, LINE 84: In humans a correlation was found between plasmatic 5-HT and the degree of hypertrophy in aortic stenosis patients. In another study, higher 5-HT levels were associated with worse HF symptoms and systolic dysfunction [19].

The authors highlighted the importance of gender on the development of depression and anxiety. However, they did not show or presented their data in terms of gender differences.

We agree with the reviewer that sex is indeed an important determinant for anxiety and depressive disorders, as well as for the development of chronic cardiac disease. According to the JCM journal guidelines, we provided a comprehensive explanation of the rationale to restrict our studies on male subjects in the methods section (experimental design). Briefly, we focused our analyses on male mice only since they exhibit more pronounced cardiac and behavioral phenotypes following myocardial infarction and decompensation to heart failure. Given that 5-HTT mutant mice of both sexes show comparable behavioral phenotypes, and the fact that females generally undergo less intense cardiac remodeling after MI, we did not include female mice in the study presented herein. The study was designed in accordance with the 3R principle, aiming to reduce the number of experimental animals to an unavoidable minimum, i.e., the reason to exclude females was also an ethical decision. As such, we are unfortunately not able to present our data in terms of sex-related differences.

To further clarify this aspect, we have adopted the limitations part of the manuscript accordingly:

PAGE 19, LINE 924: Due to established echocardiographic criteria, we could monitor CCD development in young adult male 5-HTT mice with and without experimentally induced MI, and shed light on the consequences of 5-HTT deficiency on post-MI anxiety- and depression-like behaviors by means of ethologically relevant paradigms. In the general population, however, depression is more common in women than men, and prevalence rates are also higher in female CHF patients, albeit male sex predicts poorer outcomes [84, 85]. Thus, further studies including female cohorts are warranted to decipher potential sex-specific (e.g. estrogen-driven) mechanisms regulating MI-induced cardiac remodeling and its impact on behavior due to defective 5-HTT neurotransmission.

More description on the mouse model needs to be included in the methods section.

Due to the reviewer’s suggestion, we provide an extended description on the mouse model of myocardial infarction in the methods section of the main manuscript instead of the supplement. Additionally, we adopted the methods part to provide further details on the 5-Htt mutant mouse strain.

PAGE 5, LINE 289: In brief, mice were anaesthetized with isoflurane (about 2.0 vol.%), intubated and put on a mechanical small animal ventilator. After a left-sided thoracotomy, MI was induced by permanent ligation of the proximal portion of the left anterior descending (LAD) coronary artery. Buprenorphin (0.002 mg per mouse) was given i.p. for analgesia perioperatively. In parallel subgroups thoracotomy was performed to expose the heart, but no suture was made to the coronary artery (sham operation).

PAGE 3, LINE 181: Experimental subjects were adult male mice carrying a targeted mutation of the serotonin transporter (5-Htt) gene (B6.129(Cg)-Slc6a4tm1Kpl/J; JAX stock #008355). The 5-HTT mouse line was generated as previously described [21] and had been fully backcrossed to C57BL/6J genetic background for more than 10 generations. The study cohorts comprised homozygous (5-HTT-/-) and heterozygous (5-HTT+/-) knockout mice and their corresponding wildtype (5-HTT+/+) siblings as controls. All mice were obtained from heterozygous breeding pairs, and pups were genotyped according to established protocols using ear punches to extract genomic DNA amplified by PCR. Subsequently, 5-Htt genotypes were identified by gel electrophoresis of DNA-fragments of either 225 bp (5-HTT+/+), 272 bp (5-HTT−/−) or both (5-HTT+/−) [35]. Serotonin uptake is completely abolished in homozygous mice. 5-HTT null mutants are viable and fertile, and manifest pleiotropic phenotypic traits (including neurobehavioral, metabolic and cardiovascular alterations) that emerge during early adulthood and accumulate with increasing age [45-49]. Unless otherwise stated, mice were housed in littermate groups under controlled environmental conditions (14/10 hours light/dark cycle, 21±1°C room temperature, 55±5% humidity) with standard rodent chow diet (Ssniff Spezialdiäten, Soest, Germany) and water ad libitum.

The effect of aging is not really clear in the presented experiments and the authors only chose to compare mice less than 3 months and more than 4 months, which is not enough to support their conclusions. An older age mice group at 12 10-12 months is needed to support their findings.

We agree with the reviewer that an additional middle-aged cohort of 10 to 12 months old mice would be very valuable to further deepen our research findings and to strengthen our conclusions about the age-related modulatory role of the serotonin transporter in the development and progression of cardiac pathologies. In the present study, however, we intentionally focused on young (mature) adult mice spanning approximately 2-to-6-months of age since this period of life (from early to middle adulthood) represents a particularly vulnerable time window for the occurrence and aggravation of specific phenotypic traits in 5-HTT mutant mice. For instance, 5-HTT-/- develop metabolic syndrome as they age. It is well established that mice lacking 5-HTT are prone to develop late-onset obesity (Uceyler et al. 2010, Chen et al. 2012, Zha et al. 2017, and own unpublished data) that becomes apparent in susceptible mice at approximately 4-to-6-months of age and manifests as disproportionate body weight gain, increased body fat accumulation and liver steatosis. Other intrinsic metabolic disturbances (e.g. hyperglycemia, glucose intolerance, leptin and insulin resistance) (Chen et al. 2012, Zha et al. 2017) and some behavioral abnormalities like profound hypolocomotion, however, are already detectable during early adulthood (as early as 3 months of age) (Uceyler et al. 2010, Joeyen-Waldorf et al. 2009, further already cited published data in the main text of the manuscript). Besides the specific impact of 5-HTT deficiency on the above-mentioned aging-related deteriorations, it has been reported that even minor age differences can produce significant changes in a wide range of behaviors during adulthood in nonmutant mice (Shoji et al. 2016).

These data further emphasize that the transition from early to middle life represents an extremely important time frame for studies aiming to decipher the impact of 5-HTT deficiency on development of cardiac abnormalities.

As such, we see our research design as an important starting point for more detailed future investigations and would be happy to provide further data on aging-related processes as well as the impact of 5-HTT deficiency on right ventricular function as requested by the REVIEWER 1 in upcoming studies.

To address the missing information on cardiac functioning in old 5-HTT -/- mice, we have slightly extended the methods and limitation part of the manuscript, as far as we are not able to perform these additional experiments within the scheduled review response window:

PAGE 3, LINE 190: Serotonin uptake is completely abolished in homozygous mice. 5-HTT null mutants are viable and fertile, and manifest pleiotropic phenotypic traits (including neurobehavioral, metabolic and cardiovascular alterations) that emerge during early adulthood and accumulate with increasing age [45-49].

PAGE 20, LINE 935: Besides, the 5-HT signaling system is implicated in a wide range of aging-related metabolic, cardiovascular [86] and behavioral deteriorations. As such, additional research on middle-aged (10-12-month-old) or even reproductively senescent (18-24-month-old) mice is needed to fully elucidate the modulatory role of compromised 5-HTT function on cardiac disease progression during aging.

Uceyler, N., et al., Lack of the serotonin transporter in mice reduces locomotor activity and leads to gender-dependent late onset obesity. Int J Obes (Lond), 2010. 34(4): p. 701-11.

Joeyen-Waldorf, J., N. Edgar, and E. Sibille, The roles of sex and serotonin transporter levels in age- and stress-related emotionality in mice. Brain research, 2009. 1286: p. 84-93.

Chen, X., et al., Reduced serotonin reuptake transporter (SERT) function causes insulin resistance and hepatic steatosis independent of food intake. PloS one, 2012. 7(3): p. e32511-e32511.

Zha, W., et al., Serotonin transporter deficiency drives estrogen-dependent obesity and glucose intolerance. Scientific Reports, 2017. 7(1): p. 1137.

Shoji, H., Takao, K., Hattori, S. et al. Age-related changes in behavior in C57BL/6J mice from young adulthood to middle age. Mol Brain 9, 11 (2016). https://doi.org/10.1186/s13041-016-0191-9

Were there any changes in the inflammatory mediators between the adult and the young mice?

We thank the reviewer for the valuable comment and wish to provide additional information on the age-related changes in inflammatory markers and markers of extracellular matrix.

In our baseline study, gene expression analyses were performed on myocardial tissue derived from different subsets of non-infarcted mice. Here, we refrained from a two-way full factorial analysis due to unequal sample size in the different age groups or on certain genes, respectively. However, based on the reviewer’s important remark, we re-evaluated the corresponding data by means of correlation analysis. Indeed, we found a strong positive correlation between the animals’ age and the expression level of TGF-β (age range 2.0 to 6.4 months), and an inverse correlation for Col-1α1 (age range 3.8 to 8.8 months). Expression of other relevant genes such as TNF-α or MMP-2 did not vary as a function of age, or markers were not detectable in the hearts of non-operated mice (e.g. IL-6). The observed age-related changes seemed not to be modulated by 5-Htt genotype. These findings are in line with recent literature reviewing age-related alterations in cardiac physiology and remodeling (Horn and Trafford 2016, Tominaga and Suzuki 2019, and references therein).

Based on these findings, we have expanded the results part as follows:

PAGE 9, LINE 480:  Correlational analysis indicated that the expression levels of TGF-β increased (r(40)=0.848, p<0.0001) and Col-1α1 decreased (r(24)=-0.636, p<0.001) as a function of age, though these effects were not modulated by 5-Htt genotype.

Horn MA, Trafford AW. Aging and the cardiac collagen matrix: Novel mediators of fibrotic remodelling. J Mol Cell Cardiol. 2016 Apr;93:175-85. doi: 10.1016/j.yjmcc.2015.11.005. Epub 2015 Nov 11. PMID: 26578393; PMCID: PMC4945757.Tominaga, K.; Suzuki, H.I. TGF-β Signaling in Cellular Senescence and Aging-Related Pathology. Int. J. Mol. Sci. 2019, 20, 5002. https://doi.org/10.3390/ijms20205002

It would be interesting to show the difference in altered serotonin expression between the het and the homo KO

We are pleased to provide additional information about the concentration of serotonin in mice with three different 5-Htt genotypes and adopted the introduction of the manuscript accordingly.

PAGE 2, LINE 88: Genetically modified mice lacking the serotonin transporter (5-HTT-/-) are an invaluable model to study the consequences of constitutively increased extracellular and decreased intracellular 5-HT levels due to lifelong 5-HT (re)uptake deficiency and altered 5-HT synthesis/turnover rates [18, 20-24]. Tissue concentrations of 5-HT are decreased relative to wildtype control levels to 30-60% in several brain regions and to less than 10% in multiple peripheral tissues including the heart of 5-HTT-/- mice. Conversely, neither brain nor peripheral 5-HT tissue levels are affected in mutant mice with partial loss of 5-Htt gene function (5-HTT+/-) [22]. Furthermore, 5-HTT null mutants exhibit various changes at the (neuro-)biochemical level, e.g. brain region-specific up-/downregulated expression or altered function (desensitization) of the serotonin receptors 5-HT1A, 5-HT1B, 5-HT2A and 5-HT2C that may contribute to the animals’ complex phenotypic alterations [25-30].

Round 2

Reviewer 1 Report

I thank the authors for their kind reply. The introduction of the manuscript is much improved, more concise and informative. The authors' greater caution in reporting a causal relationship between inflammation and deficit in the healing process in  5-HTT knockout mice is also appreciable. However, many of the major concerns I have reported have not been resolved in this new version of the manuscript. I understand the difficulty in performing experiments on a new cohort of animals (in terms of timing). However, I believe that it is still possible to work on the specimen and on the functional data already collected. In fact, as noted in my previous comments, the hemodynamic data need revision and some crucial independent afterload measures (i.e. ESPVR, PRSW) are not shown. I recommend the authors to work on the material already collected to shed light on a cardiac phenotype that still appears too nebulous.

Author Response

Answer to Review 1 comments / jcm-1266884

Reviewer #1:

FIRST REVISION ROUND:

The manuscript entitled 5-HTT deficiency in male mice affects healing and behavior after myocardial infarction is characterized by interesting and useful insights for understanding how serotonin and its transporter SERT are able not only to influence responses at central level, but also at cardiac one.

The work is well designed from an experimental point of view and abounds with many research methodologies that enrich its content.

However, the research has some limitations that in my opinion need to be overcome.

Despite the massive amount of experiments performed, it remains to be determined why naïve 5-HTT-/- mice are more prone to develop age-dependent cardiomyopathy and heart failure. Echocardiographic and hemodynamic data are shown to support the thesis, however as the authors pointed out, no LV fibrosis, structural alterations and remodeling is present in the heart of the 5-HTT-/- mice. Furthermore, the hemodynamic data do not clearly support the thesis of an intrinsic myocardial impairment, indeed if it’s true that CO (measure not load independent and affected by HR that is diminished in the mutants) is decreased, dP/dt max is totally unchanged and what is more no measures afterload independent are shown (i.e. ESPVR and PRSW). Conversely the most affected parameter among the hemodynamic ones, P at dV/dt max (with an enormous increase observed in the total KO mice), seems to indicate an afterload problem (supported also by the trend observed in EA).

In light of this, I advise the authors to delve into the role of systemic blood pressure. Indeed, many works suggest a crucial role of serotonin (Watts et al. 2012) and of the transporter SERT (Patrick Davis et al. 2011) in regulating systemic blood pressure. A role of hypertension could also explain the enormous increase in mortality after MI, with no differences in the infarcted area observed among genotypes.

SECOND REVISION ROUND:

I thank the authors for their kind reply. The introduction of the manuscript is much improved, more concise and informative. The authors' greater caution in reporting a causal relationship between inflammation and deficit in the healing process in  5-HTT knockout mice is also appreciable. However, many of the major concerns I have reported have not been resolved in this new version of the manuscript. I understand the difficulty in performing experiments on a new cohort of animals (in terms of timing). However, I believe that it is still possible to work on the specimen and on the functional data already collected. In fact, as noted in my previous comments, the hemodynamic data need revision and some crucial independent afterload measures (i.e. ESPVR, PRSW) are not shown. I recommend the authors to work on the material already collected to shed light on a cardiac phenotype that still appears too nebulous.

We thank the reviewer for the important feedback regarding the importance of hemodynamics in mice model of the 5-HTT deficiency. Recently, we have routinely started to measure afterload independent parameters (i.e. ESPVR, PRSW as proposed by the reviewer). We looked at our original loops, but the compression of vena cava to assess afterload independent parameters has unfortunately not been routinely assessed at the time of 5-HTT experiments. However, due to the further analysis of the original loop recordings and original hemodynamics data, we are now able to provide information on both systolic and diastolic blood pressure in baseline assessment of mice with three different 5-Htt genotypes as advised by the reviewer in the first version of the reviewer comments. Both the systolic (p=0.990) and the diastolic (p=0.818) blood pressure did not differ within the mice with three genotypes: 5-HTT+/+ (101.3 ± 7.1 mmHg and 64.1 ± 4.1 mmHg), 5-HTT +/- (100.4 ± 5.3 mmHg and 61.8 ± 3.8 mmHg) and 5-HTT -/- (101.4 ± 4.2 mmHg and 60.6 ± 2.4 mmHg), respectively. We have added this information both in the manuscript and in the Supplemental table S2 (PAGE 6 of the Supplement). The development of both hypo- and hypertension after 5-HT application has been already further dissected in the introduction part of the manuscript as part of the first revision.

PAGE 8, LINE 399: No differences could be obtained regarding systolic and diastolic blood pressure between the different genotypes.

Furthermore, we have addressed the concerns of the reviewer regarding the definition of “cardiomyopathy” in 5-HTT deficient mice and adopted the wording accordingly in the manuscript. We propose now to describe the changes found in 5-HTT -/- mice as “cardiac dysfunction”.

PAGE 1, LINE 27: Cardiological evaluation of experimentally naïve male mice revealed a mild cardiac dysfunction dilated cardiomyopathy-like phenotype in ≥ 4-month-old 5-HTT knockout (-/-) animals.

PAGE 1, LINE 38: This study shows that 5-HTT deficiency leads to age-dependent cardiomyopathy cardiac dysfunction and disrupted early healing after MI probably due to alterations of inflammatory processes in mice.

PAGE 9, LINE 409: Age-dependent development of heart failure phenotype cardiac dysfunction in 5-HTT deficient (-/-) mice.

PAGE 20, LINE 901: This study shows that 5-HTT deficiency leads to age-dependent cardiomyopathy cardiac dysfunction and disrupted early healing after myocardial infarction, which was associated with most likely due to increased local inflammation and MMP-2 expression during the early stage after MI.

Reviewer 2 Report

The authors responded appropriately and very thoroughly to the previous comments and concerns.

Author Response

The reviewer 2 had no further concerns regarding the manuscript.